# 3D microprinting of inorganic porous materials by chemical linking-induced solidification of nanocrystals

Minju Song [1], Yoonkyum Kim[1], Du San Baek [2], Ho Young Kim[3], Da Hwi Gu[1], Haiyang Li[4], Benjamin V. Cunning[5], Seong Eun Yang[1], Seung Hwae Heo[4], Seunghyun Lee[1], Minhyuk Kim[6], June Sung Lim [7,8], Hu Young Jeong [6], Jung-Woo Yoo [1], Sang Hoon Joo [8], Rodney S. Ruoff [1,2,5], Jin Young Kim [3] ✉ & Jae Sung Son [4] ✉

Three-dimensional (3D) microprinting is considered a next-generation manufacturing process for the production of microscale components; however, the narrow range of suitable materials, which include mainly polymers, is a critical issue that limits the application of this process to functional inorganic materials. Herein, we develop a generalised microscale 3D printing method for the production of purely inorganic nanocrystal-based porous materials. Our process is designed to solidify all-inorganic nanocrystals via immediate dispersibility control and surface linking-induced interconnection in the nonsolvent linker bath and thereby creates multibranched gel networks. The process works with various inorganic materials, including metals, semiconductors, magnets, oxides, and multi-materials, not requiring organic binders or stereolithographic equipment. Filaments with a diameter of sub-10 μm are printed into designed complex 3D microarchitectures, which exhibit full nanocrystal functionality and high specific surface areas as well as hierarchical porous structures. This approach provides the platform technology for designing functional inorganics-based porous materials.

Additive manufacturing, commonly known as three-dimensional (3D) printing technology, allows the production of materials with customised shapes and dimensions[1,2]. In particular, micro-stereolithography techniques[3,4] and direct ink writing[5] can revolutionise the manufacturing of microscale components for various applications, including micromechanics, microelectronics, and biomedical systems by enabling the creation of previously inaccessible 3D architectures[6–10].

Micro-stereolithography is based on multiphoton absorption and is achieved using a projection lens, while direct ink writing employs microneedles; however, these processes critically rely on organic or polymer-based resin inks to ensure photocurability or rheological printability in the optical lithography or ink writing process, respectively, which limit the range of printable materials and the intrinsic functionality of the printed objects. Nanomaterial-based polymer

[1]Department of Materials Science and Engineering, Ulsan National Institute of Science and Technology (UNIST), Ulsan 44919, Republic of Korea. [2]Department of Chemistry, Ulsan National Institute of Science and Technology (UNIST), Ulsan 44919, Republic of Korea. [3]Hydrogen·Fuel Cell Research Center, Korea Institute of Science and Technology (KIST), 14-gil 5 Hwarang-ro, Seongbuk-gu, Seoul 02792, Republic of Korea. [4]Department of Chemical Engineering, Pohang University of Science and Technology (POSTECH), Gyeongsangbuk-do 37673, Republic of Korea. [5]Center for Multidimensional Carbon Materials (CMCM), Institute for Basic Science (IBS), Ulsan 44919, Republic of Korea. [6]Graduate School of Semiconductor Materials and Devices Engineering, Ulsan National Institute of Science and Technology (UNIST), Ulsan 44919, Republic of Korea. [7]School of Energy and Chemical Engineering, Ulsan National Institute of Science and Technology (UNIST), Ulsan 44919, Republic of Korea. [8]Department of Chemistry, Seoul National University, Seoul 08826, Republic of Korea. ✉e-mail: jinykim@kist.re.kr; sonjs@postech.ac.kr

resin-free printing techniques were recently developed by introducing photocurable molecules to the nanomaterial surfaces[11,12]. Sun et al. achieved the stereolithographic 3D nanoprinting of quantum dots by exploiting photoexcitation-driven chemical bonding[12]. Despite these recent successes, existing processes for inorganic nanomaterials or composites are only applicable to specific photo-active materials and require special lithographic equipment. Thus, new processes are required to diversify and expand the range of inorganic materials suitable for 3D microprinting.

Colloidal nanocrystals have emerged as versatile inorganic building blocks of functional 2D and 3D solids with tailorable physicochemical properties[13–15], while advances in synthetic methodology have enabled the preparation of colloidal particles of almost any inorganic functional material[16–18]. In particular, the assembly of colloidal nanocrystals into macroscopic solid gels introduces a new class of inorganic porous material with high surface areas and low densities[19,20], whose electronic, magnetic, and optical properties originate from building blocks[21–23], and can thus be tailored to a wide variety of applications, including energy storage and conversion[24,25], catalysis[26,27], adsorbent[28], filter[29], and electrodes[30], and sensors[31]. Herein, we demonstrate a generalised method to achieve high-resolution wet 3D microprinting of inorganic porous materials by the direct writing of purely inorganic colloidal nanocrystal ink in the linker-containing nonsolvent bath (Fig. 1a). This process is applicable to a diverse range of materials and produces crystalline nanostructures with high structural integrity. Further, the process is tailored toward the creation of purely inorganic materials with 3D microarchitectures and does not require any polymer resins. The inorganic ligand-capped nanocrystal inks were solidified by the instant interconnection of the nanocrystals through solvent polarity change and surface linking-induced interconnection during the ink extrusion, which affords a multibranched gel network (Fig. 1b). The dimension-controlled printing of microscale filaments creates complex 3D architectures, which retain the functionalities of the primary nanocrystals and exhibit high specific surface areas comparable to those of existing nanocrystal-based and sol-gel-processed aerogels, promoting material functionalities. Further, our method produced multiple materials using mixed nanocrystal inks or sequential printing.

## Results

### Wet 3D microprinting of inorganic nanocrystals

The developed wet 3D microprinting method involves three steps: (i) preparation of a negatively charged nanocrystal inks, (ii) controlled solidification of the extruded nanocrystals in the nonsolvent bath containing linker ions, and (iii) supercritical drying of the wet-state of printed objects into the solid-state 3D architectures. Inorganic ligand (i.e., tetrathiomolybdate; $MoS_4^{2-}$)-capped nanocrystals dispersed in polar solvent (i.e., N-methylformamide (NMF)) were exploited as printing inks, which are extruded into nonsolvent with lower dielectric constant (i.e., butanol) containing linker ions. Instantaneous solvent-to-nonsolvent mixing during the ink extrusion instantly reduces the dispersibility of negatively charged nanocrystals, resulting in agglomeration of nanocrystals. Systematic studies on the combination of solvent (NMF) and nonsolvent with respect to their solubility parameters and dielectric constants reveal that the optimum ranges are essential to flocculate nanocrystals (Supplementary Fig. 1 and Supplementary Table 1). When the differences in solubility parameters

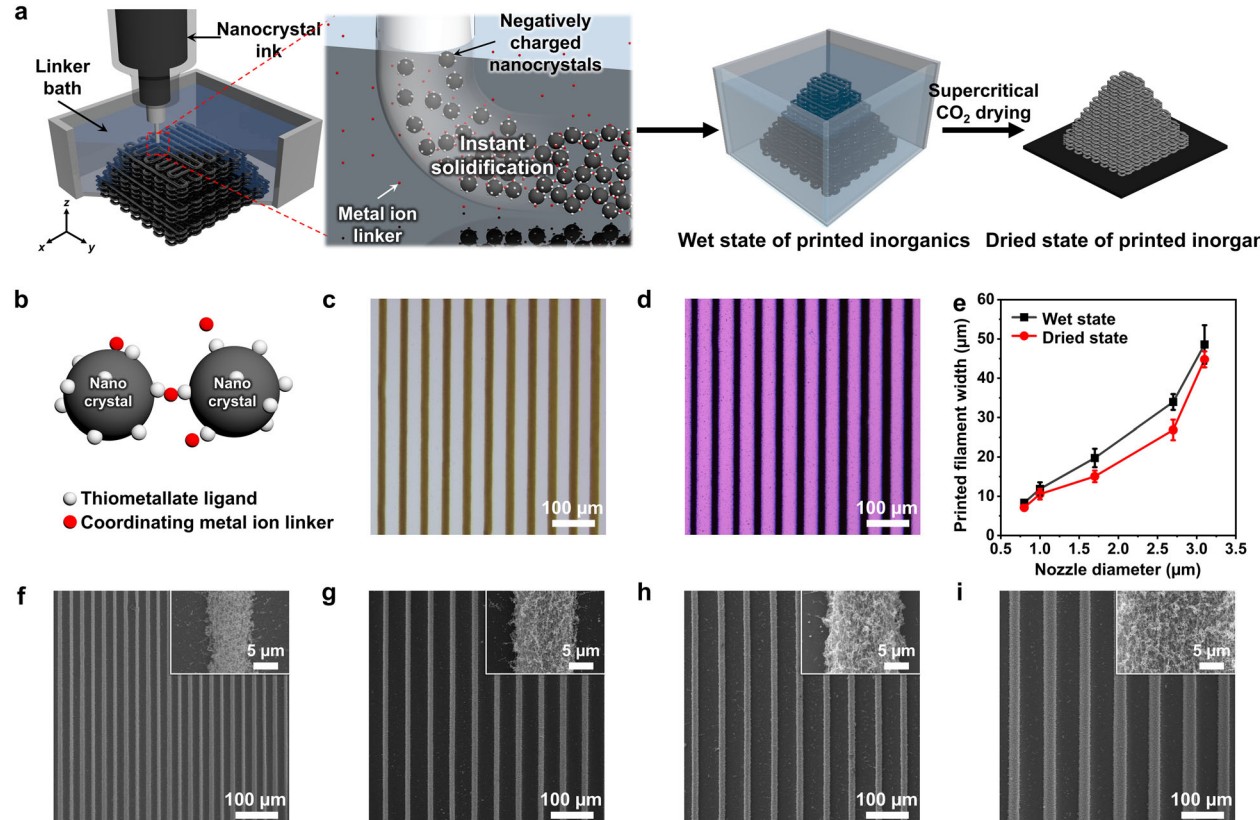

**Fig. 1 | 3D microprinting of inorganic nanocrystals. a** Scheme for wet 3D microprinting of inorganic nanocrystal-based porous materials. **b** Schematic illustration of linking of all-inorganic nanocrystals with coordinating metal ions. The components are colour-coded as follows: grey, nanocrystal; white, thiometallate anion ligand; red, metal ion linker. OM images in wet state (**c**) and dried state (**d**) of printed multilayer Ag filaments. **e** Widths of printed Ag filaments as a function of nozzle diameter for the wet state and dried state. The error bars represent the standard deviation of the 15 lines of printed Ag filament widths. SEM images of multilayer Ag filaments patterned with feature widths of 7- (**f**), 11- (**g**), 15- (**h**), and 27-μm (**i**). Insets: high magnification SEM images of corresponding filaments.

of nonsolvent with NMF exceeded certain limits, their phases are separated. Also, the nonsolvent with high dielectric constant solubilized the negatively charged nanocrystals. Under the optimum ranges of both characteristics of nonsolvent, the instant agglomeration of nanocrystals was achieved (Supplementary Figs. 2, 3 and Supplementary Table 1).

Multivalent metal ion linkers play the role of creating covalent bonds with the surface inorganic ligands among nanocrystals in agglomerates which enhance the overall structural entanglement (Supplementary Fig. 4), eventually leading to the robust solidification. Since the thiomolybdate anion ligand (i.e., a soft Lewis base) has chemical affinity to metal ion linkers that act as soft Lewis acids[32,33], soft Lewis acids of multivalent coordinating ions, including $Au^{3+}$, $Pt^{4+}$, and $Fe^{2+}$ but not limited, were chosen to serve as metal linkers (e.g., $Au^{3+}$ linker for Au and Ag nanocrystals, $Pt^{4+}$ for FePt nanocrystals, and $Fe^{2+}$ for CdSe and $Fe_3O_4$ nanocrystals). The solidification of the nanocrystals was optimised using the control parameters of the concentrations of both the metal ion linkers and the nanocrystals. When the concentrations of either the nanocrystals or the coordinating linker ions exceeded certain limits, the nozzle was easily clogged, causing discontinuous ink deposition (Supplementary Fig. 5). In contrast, decreasing their concentrations "too far" caused the dissolution of nanocrystals into the linker bath, and we could not obtain the defined filaments by printing (Supplementary Fig. 5). In turn, under optimum conditions, defined 3D inorganic filaments with multibranched porous nanocrystal networks were printed by our process.

A model system consisting of thiomolybdate-capped Ag nanocrystal building blocks was studied. The electrokinetic $\xi$-potentials changed from −50.7 to −10.1 mV during the deposition, indicating that the nanocrystals lost their surface charges upon coordination of the metal ion linkers (Supplementary Fig. 6). Moreover, the peaks in the S 2p region of the X-ray photoelectron spectroscopy (XPS) spectrum showed a peak shift to higher binding energies upon ligation to metal ions, while the Ag 3d peaks did not change (Supplementary Fig. 7). This result confirms that the nanocrystals were bridged by linking the surface thiometallate ligands with metal ions.

The system used to print the inorganic nanocrystals consisted of a micropipette nozzle connected to a syringe-type reservoir containing negatively charged inorganic nanocrystals, which were extruded using a pneumatic pressure controller. The stage mounted with the linker-containing solidification bath was moved along the x-, y-, and z-axes in a pre-designed model (Supplementary Fig. 8). The diameters of the filaments were precisely controlled from 8 to 49 μm in the wet state by adjusting the nozzle diameter; thus, the method demonstrates microscale printability. The wet filaments were further dried by supercritical $CO_2$ drying, effectively generating extremely microporous materials without substantial structural shrinkage or distortion, as shown in the optical microscopy (OM) images (Fig. 1c, d). Scanning electron microscopy (SEM) images (Fig. 1f–i) confirmed that the dried filaments constructed from Ag nanocrystals showed uniform linewidths with controllable diameters ranging from 7 to 44 μm (Fig. 1e). The average linewidths of the Ag filaments were estimated from 15 filaments in the SEM images of each sample. High-resolution (HR) SEM images clearly revealed extremely porous and multibranched networks in all samples, which showed characteristics of typical aerogels (Fig. 1f–i, insets).

This ability to print microscale inorganic filaments enables the construction of complex 3D architectures in diverse ranges of materials. For example, a 3D cubic lattice structure was built via the layer-by-layer deposition of 32 Ag filament layers in a single pass (Fig. 2a, Supplementary Fig. 9, Supplementary Movie 1). The printing method showed high precision, while the printed 3D architectures exhibited excellent structural fidelity and were consistent with the design model (Fig. 2a, inset). The 3D-printed filaments maintained a circular cross section with a uniform diameter (Fig. 2b, c). Meanwhile, the filaments became flattened when they are stacked to the multi-layered structures due to the mechanical plasticity of wet-gels (Supplementary Fig. 9). Energy dispersive X-ray spectroscopy (EDS) maps showed that Ag nanocrystals, S-based ligands, and Au-based linkers were confined to the patterned layers (Fig. 2d). Lattice structures were printed using various materials, including metallic Au, magnetic FePt and $Fe_3O_4$, and semiconducting CdSe nanocrystals (Fig. 2e–h). SEM images and EDS maps confirm that all samples show high structural and compositional integrity and uniformity (Supplementary Fig. 10). Various 3D objects were printed, including a CdSe-based pyramid (Fig. 2i, Supplementary Movie 2) and a FePt-based hexagonal prism (Fig. 2j, Supplementary Movie 3), which all showed excellent lateral and vertical shape fidelity. Moreover, a large square lattice pattern with dimensions of 1.2 mm × 1.2 mm was constructed by the layer-by-layer printing of 8 Ag filament layers (Fig. 2k), thereby demonstrating the feasibility of the printing process over a millimetre-scale area. These examples of complex 3D architectures built from various nanocrystals clearly demonstrate the microscale 3D printability of the method and the wide range of applicable materials.

## Microstructural characteristics

Low-resolution and HRSEM images of the printed structures reveal porous microstructures with interconnected networks of extremely thin wire-like structures with numerous bifurcations (Fig. 3a, b). The typical dimensions of these multibranched wire-like structures are on the same size scale as the diameter of the original nanocrystals (5–10 nm), which demonstrates that the printed structures were formed directly from the original colloidal nanocrystal building blocks without the formation of any kind of secondary structures. A diverse range of pore sizes was observed, ranging from a few nanometres to several hundreds of nanometres. TEM images (Fig. 3c, Supplementary Fig. 11) show that the nanocrystals were maintained their primary shapes and sizes. HRTEM images of the printed Au nanocrystals (Supplementary Fig. 12) reveal that the printed Au nanocrystals were touched at the junction in which the neck regions are slightly broadened. However, the substantial merging was not observed and all individual nanocrystals maintained their primary nanostructural characteristics. The printed FePt nanocrystals have similar characteristics in HRTEM image, in which the nanocrystals were well isolated or slightly touched at the junctions. The random orientation of the nanocrystals in the chains resulted in ring-like electron diffraction patterns (Fig. 3d). X-ray diffraction (XRD) spectra of the as-synthesised nanocrystals and printed structures were almost identical in terms of the observed peak widths and positions (Fig. 3e, Supplementary Fig. 13). The calculated crystallite sizes of the printed nanocrystals from the XRD patterns with the Scherrer formula (Supplementary Table 2) were similar or slightly larger than those of as-synthesised ones. This result indicates that the particles are not merged into larger aggregates, further suggesting the near-perfect persistence of the nanocrystals through the entire process. Thermogravimetric analysis of the 3D-printed Au showed total weight losses of less than 5%, even at temperatures as high as 700 °C (Supplementary Fig. 14), considerably lower than the 20–30% weight loss typically observed in organic-capped nanocrystal-based aerogels. This result further illustrates the stability of the printed inorganic aerogels.

We conducted microstructural analyses on both internal and external structures of the printed filaments with different diameters since one would expect for large diameter structures, the 'curing' of the outer layer limits transport of the nonsolvent into the interior of the printed filament. However, in all samples, no apparent differences in microstructures including porosity and fractal structures were observed. Similarly, the outer surface of the filament displayed microstructures consistent with the corresponding internal structures. These results strongly suggest the homogeneity of microstructures in the printed filaments, irrespective of their diameters. The microstructural analysis on the filament with different diameters

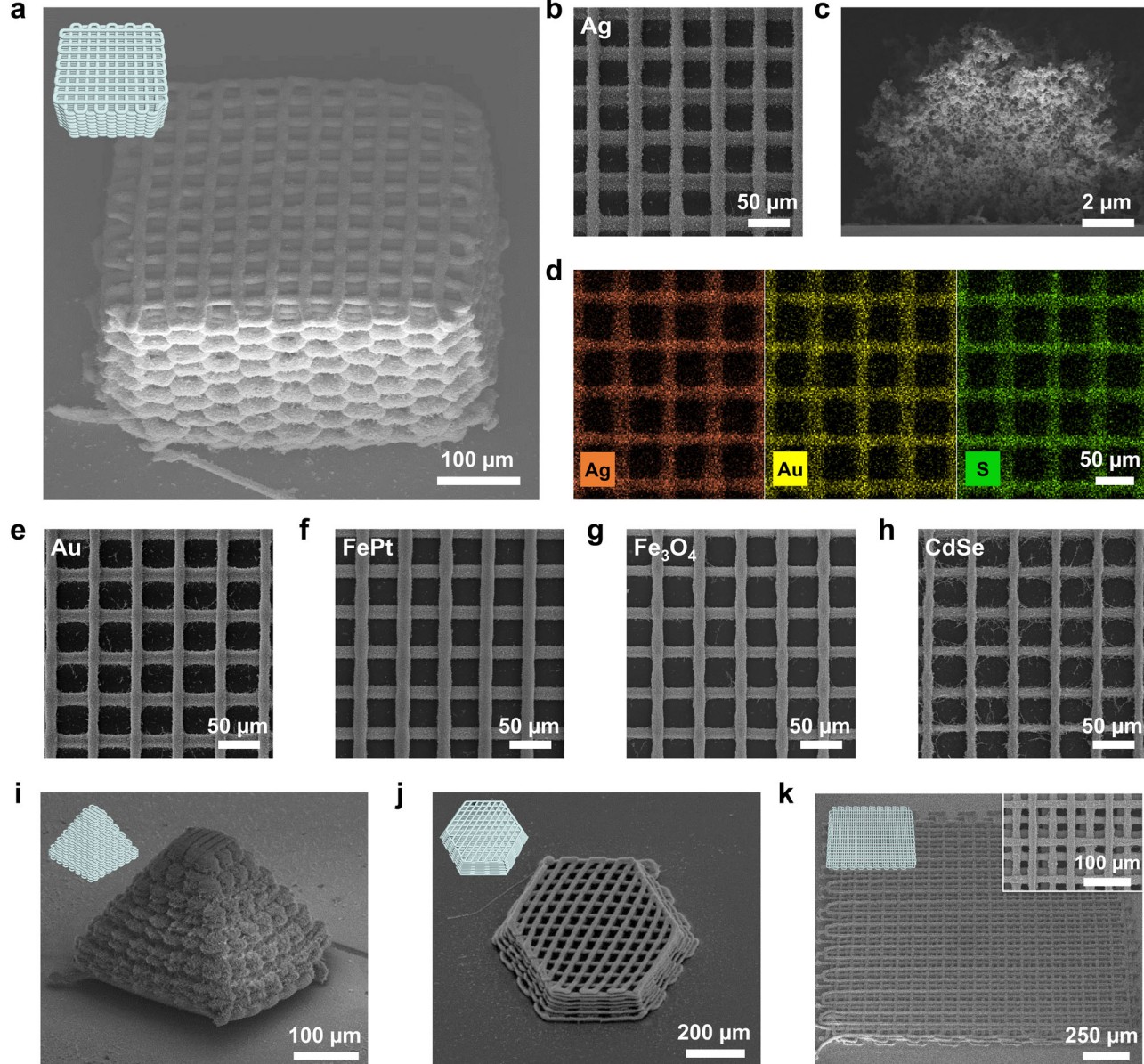

**Fig. 2 | Printed 3D microarchitectures of inorganic porous materials. a** 32-layer lattice cube illustration model (inset) and SEM image of 3D-printed Ag. **b** Top view of SEM image of 3D-printed Ag lattice cube. **c** Cross-sectional SEM image of printed Ag filament. **d** EDS mapping image of printed Ag comprising Ag nanocrystals (orange), Au-based linkers (yellow) and S-based ligands (green). SEM images of 3D lattice structures of Au (**e**), FePt (**f**), $Fe_3O_4$ (**g**), and CdSe (**h**). **i** 36-layer pyramid illustration model (inset) and SEM image of 3D-printed CdSe. **j** 12-layer hexagonal prism illustration model (inset) and SEM image of 3D-printed FePt. **k** 8-layer large-scale square lattice structure illustration (left inset) and SEM image of 3D-printed Ag. Right inset: magnified SEM image of printed structure.

was further discussed in the Supplementary Discussion (Supplementary Figs. 15–17 and Supplementary Table 3).

To further investigate the interplay between reaction and diffusion for curing nanocrystals, the microstructural analyses were performed for the Ag filaments printed with different nonsolvents in the solidification bath, such as ethanol, 2-propanol, 1-butanol, 1-pentanol, and 1-hexanol. By employing these solvents, we were able to systematically examined the impact of nanocrystal solubility changes (i.e., dielectric constants) when combined with the solvent, NMF. Particularly noteworthy is the fact that these alcoholic solvents exhibit varying diffusion coefficients, with a decrease observed in alcohols possessing longer hydrocarbon chains, facilitating the precise observation of the diffusion dynamics of the nonsolvent during the printing process[34]. With increasing the hydrocarbon chains of alcoholic nonsolvents, the printed filaments became narrowed (Supplementary Fig. 18a–c). This

intriguing observation could be attributed to alterations in the solubility of nanocrystals, where the use of bulkier nonsolvents with lower dielectric constants led to reduced solubility of the nanocrystals, thereby limiting the outward diffusion of nanocrystals towards the nonsolvent region and ultimately resulting in narrower filaments. Slower inter-diffusion between NMF and bulkier nonsolvent can also contribute to limit the outward diffusion of nanocrystals. Moreover, owing to the narrower dimension of the filament printed with a bulkier nonsolvent, they exhibited denser microstructures and smaller pore sizes (Supplementary Fig. 18f–o).

To gain a quantitative understanding of the nonsolvent effect, the mass fraction dimensions ($d_f$) were calculated from the TEM images of the corresponding fractal structures. The mass fractal dimension is a well-established parameter utilised for characterising fractal structures, including aggregates and agglomerates resulting from

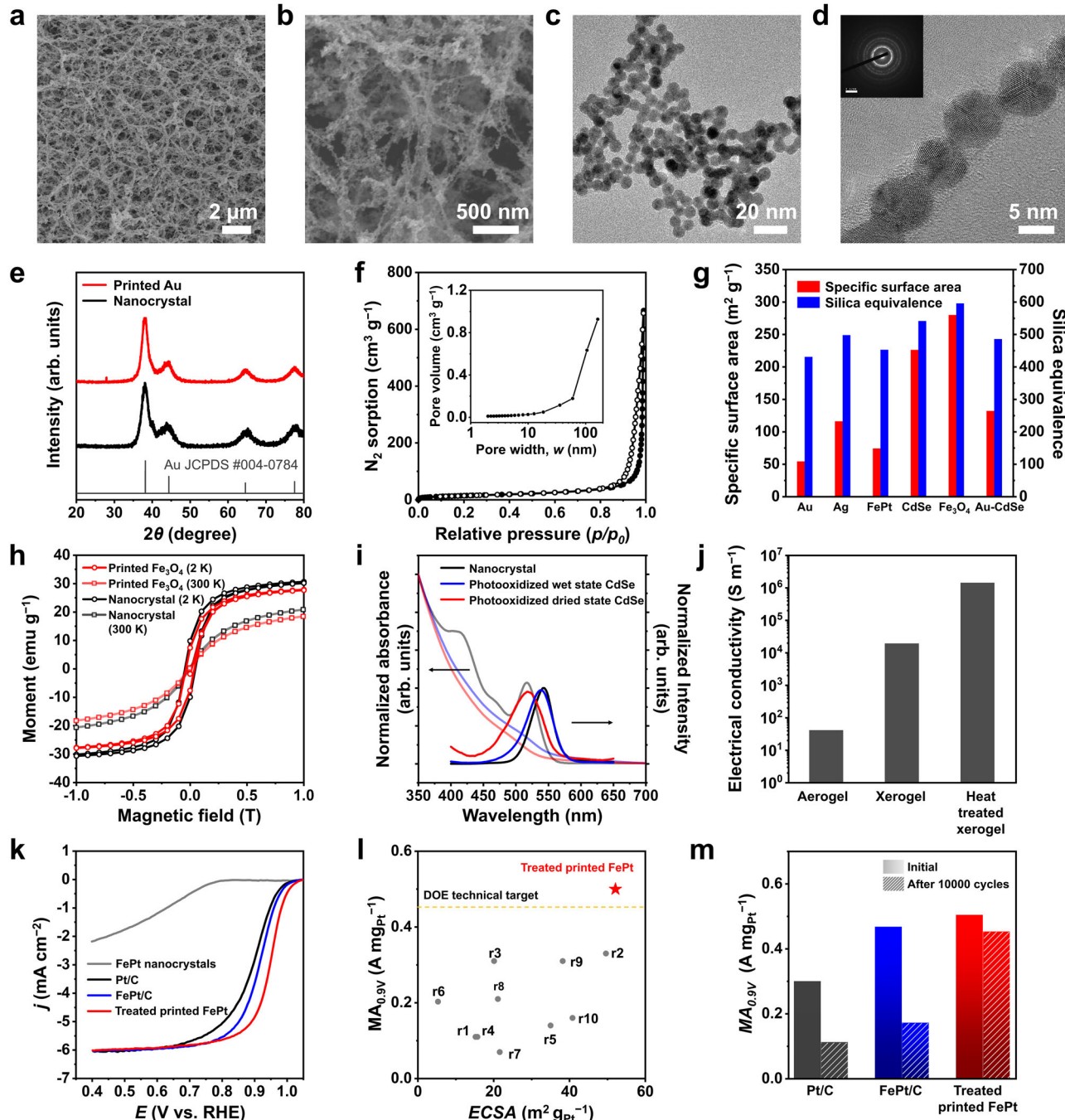

**Fig. 3 | Microstructural characteristics and functionalities of the 3D-printed materials. a, b** HRSEM images of printed Au nanocrystals. TEM (**c**) and HRTEM (**d**) images of printed Au nanocrystals. Inset in the (**d**): electron diffraction pattern. **e** XRD patterns of as-synthesised Au and printed Au nanocrystals. **f** N₂ physisorption isotherms of printed Au sample at 77 K. Inset: pore size distribution derived from Barrett-Joyner-Halenda (BJH) analysis. **g** Summarised specific surface areas and silica equivalents of printed inorganic porous materials. **h** Magnetic hysteresis loops of as-synthesised $Fe_3O_4$ (black) and printed nanocrystals (red) measured at 2 K and 300 K. **i** UV-Vis absorption and PL spectrum of CdSe nanocrystals (black) and printed wet (blue), and dried (red) samples. **j** Electrical conductivities of printed Au porous sample, xerogel, and sintered sample. **k** ORR polarisation curves of FePt nanocrystal building blocks, Pt/C, FePt/C, and electrochemically activated, HCl-treated, printed FePt nanocrystals (treated printed FePt). The catalyst loading amounts were optimised at 20 μg$_{Pt}$ cm⁻² for Pt/C and FePt/C and at 50 μg$_{Pt}$ cm⁻² for treated printed FePt. **l** ECSA vs. MA$_{0.9\,V}$ plot for the comparison of this work with the previous literatures of Pt-based self-supported ORR catalysts. US Department of Energy (DOE) technical target for ORR MA$_{0.9\,V}$ (0.44 A mg$_{Pt}$⁻¹) are presented as yellow line. **m** MA$_{0.9\,V}$ values before and after the 10,000 cycles of ORR accelerated durability test (ADT) in 0.1 M HClO₄.

colloidal suspensions of particles[35]. In particular, the $d_f$ serves as a quantitative tool for comprehending the formation mechanism of aggregates, particularly in scenarios where the interplay between diffusion and reaction of particles plays a pivotal role. As depicted in Supplementary Fig. 19 and Table 4, the $d_f$ gradually increases as the hydrocarbon chains lengthen. This trend indicates that the

solidification of nanocrystals becomes more reaction-limited in the bulkier nonsolvent bath, rather than being limited by the diffusion of nanocrystals toward each other. Accordingly, it can be inferred that the bulkier nonsolvents with lower diffusion coefficients can be slowly exposed to the deposited nanocrystals in inks, leading to a slower solidification reaction. In other words, the non-solvent diffusion rate is

a determinant of the solidification reaction kinetics in the printing of nanocrystals and consequently affects the microstructure. These findings show the significant impact of the fundamental diffusion and reaction kinetics on both the macroscopic dimensions and microstructural characteristics of the printed filaments.

The specific surface area and porosity of the printed structures were evaluated using nitrogen adsorption/desorption isotherms. All samples exhibited typical type II isotherms with type H3 hysteresis loops similar to those of known nanocrystal-based aerogels at high relative pressures (Fig. 3f, Supplementary Fig. 20). The pore size distribution plots of the printed structures confirm the broad pore size distribution (almost independent of the materials), indicating that the formation of pores was induced by the interconnection of nanocrystals and was therefore independent of the characteristics of the nanocrystal cores (Fig. 3f inset, Supplementary Fig. 21). The printed $Fe_3O_4$ and Au samples showed the highest (279 $m^2 g^{-1}$) and lowest (54.2 $m^2 g^{-1}$) Brunauer-Emmett-Teller (BET) surface areas, respectively, which is in good agreement with those of reported nanocrystal-based aerogels. The mole-based surface areas of the printed materials are very high because the molecular weights of these materials are significantly higher than those of well-known silica aerogels. The silica-equivalent surface areas of the 3D-printed structures ranged from ~400 to ~600 $m^2 g^{-1}$ (Fig. 3g, Supplementary Table 5), which are comparable to those of nanocrystal- and oxide-based aerogels[36–43]. In addition, the printed Ag samples with bulkier nonsolvents exhibited lower surface areas (Supplementary Fig. 21 and Table 5), agreeing with the microstructure analysis results (Supplementary Fig. 18).

## Diverse functionality of printed materials

The solidification of the inorganic nanocrystals induced by the surface linking-induced interconnection effectively conserved the primary functionalities of the nanocrystals in the 3D-printed structures. For example, the 3D-printed $Fe_3O_4$ exhibited superparamagnetic properties like those of the as-synthesised $Fe_3O_4$ nanocrystals, with a blocking temperature of 28 K, and in good agreement with the preserved nanostructural characteristics observed in the SEM and TEM images (Supplementary Fig. 22). The magnetisation curves also confirm the conservation of the magnetic properties of $Fe_3O_4$ nanocrystals, showing that the saturation magnetisation and coercive fields of printed and as-synthesised nanocrystals at 2 K and 300 K were essentially identical (Fig. 3h). As another example, the 3D-printed wet and dried CdSe structures retained the excitonic features of the CdSe semiconductor nanocrystals. Even though a small red-shift and peak broadening were observed in the absorption spectrum owing to electronic coupling among nanocrystals in the connected networks, the printed CdSe still exhibit a relatively sharp onset, indicating that the quantum confinement effect of the nanocrystals is retained in the printed structure (Supplementary Fig. 23). The luminescent properties of the printed CdSe were recovered by the photooxidation treatment, although the luminescent properties of the inorganic CdSe nanocrystals in the ink were initially quenched via the surface exchange with the inorganic thiometallate ligands, which introduced numerous surface traps that cause non-radiative recombination. Our previous work described that the photooxidation of inorganic ligand-capped CdSe nanocrystals enhances their luminescent properties by the passivation of surface dangling bonds via surface oxidation[44]. The printed CdSe exhibits relatively sharp emission along with a small blue-shift and peak broadening in the photoluminescent (PL) spectrum (Fig. 3i). This is coincident with the absorption spectrum, which shows the peak shift and broadening, indicating that the PL properties originate from the band edges rather than the surface trap states. The blue shifts observed in the CdSe solid were attributed to deep surface oxidation, which reduced the size of the CdSe core.

As a third example, the electronic coupling between metal nanocrystals in the percolating porous structure was responsible for its favourable electrical transport properties (Fig. 3j, Supplementary Fig. 23). The 3D-printed and $CO_2$-dried Au sample exhibited an electrical conductivity of $4.0 \times 10^1$ S $m^{-1}$, which is within in the reported range of the reported conductivities of porous metal materials[45]. Drying the wet Au sample under ambient conditions induced a transformation into a denser xerogel with a significantly enhanced electrical conductivity of $1.9 \times 10^3$ S $m^{-1}$. Heat treatment at 600 °C further increased the electrical conductivity to $1.4 \times 10^6$ S $m^{-1}$, which is within the range of the conductivities of micro-patterned electrodes in microelectronic systems[46]. These results agree with their microstructures, where the macroscopic porosity was significantly reduced in the dried xerogel and heat-treated xerogel, compared with the aerogel (Supplementary Fig. 25a–c). In particular, the HRTEM images (Supplementary Fig. 25d–i) revealed that the nanostructural microstructures were preserved in the xerogel, while the sintered samples exhibited remarkably larger grains, indicative of effective sintering during the heat treatment process. These microstructural changes were further confirmed through the XRD patterns of the samples (Supplementary Fig. 25g), wherein the sintered sample demonstrated notably sharpened peaks. In the EDS elemental analysis of the aerogel, only Au nanocrystals and surface ligand (Mo and S) were detected without any impurities. Notably, in the heat-treated sample, the proportion of Mo significantly reduced (Supplementary Table 6). This result indicates the effectiveness of ambient drying accompanied with washing with solvent and heat treatment for removal of the $MoS_4^{2-}$ thiometallate ligand, as this molecule is known to thermally decompose at 200–500 °C[47].

Another important functionality of porous noble metals pertains to their huge surface areas that promote chemical and electrochemical catalytic activities. Our printed structures manifest a pore hierarchy encompassing macro-, meso-, and micropores across their entire domain, distinguishing them from earlier 3D thin-film catalysts[48–50]. This distinction arises from preserving the structure of nanocrystal building blocks and interstitial micropores throughout the fabrication procedures. This hierarchical porosity can enhance molecular accessibility and the even distribution of three-phase interfaces, promoting reaction kinetics and boosting catalytic activity[51]. To investigate the electrocatalytic activity of our printed material with high porosity, the printed FePt porous material was chosen for the oxygen reduction reaction (ORR) because the FePt nanostructures are renowned as highly active catalysts with a suitable binding energy between their surfaces and the ORR intermediate (Supplementary Fig. 26)[52]. To afford clean surfaces for promoting the catalytic activity, we removed the metal ion linker and surface ligands in printed materials via acid treatment in 0.1 M HCl solution and subsequent electrochemical cycling activation in 1.0–1.5 V (vs. RHE) (Supplementary Fig. 27 and Table 7)[53,54]. Unlike the untreated sample, the treated sample clearly exhibited high ORR activity with higher half-wave potential ($E_{1/2}$) at 0.947 V than those of commercial Pt/C (0.900 V) as well as FePt/C (0.914 V) with the same composition (Fig. 3k). Importantly, the electrochemically active surface area (ECSA) and ORR mass activity at 0.9 V (vs. RHE) ($MA_{0.9V}$) of the treated sample surpass those of Pt-based self-supported ORR catalysts reported to date (Fig. 3l; Supplementary Figs. 28–29 and Table 8). Moreover, the 3D interconnected contiguous nanostructure of printed materials mitigated the dissolution and agglomeration of individual nanocrystals, enabling excellent ORR durability of the treated sample[55]. After the 10,000 accelerated durability test (ADT) cycles, the treated sample retained 89.8% of its initial $MA_{0.9V}$, while FePt/C and Pt/C preserved only 36.7 and 37.2%, respectively (Fig. 3m, Supplementary Figs. 30–31). The electrocatalytic activity was further discussed in the Supplementary discussion.

## 3D printing of multi-material architectures

Porous structures with multiple material compositions were 3D-printed by sequentially printing different nanocrystal inks, or by printing mixed inks containing different nanocrystals. A combination of Au and

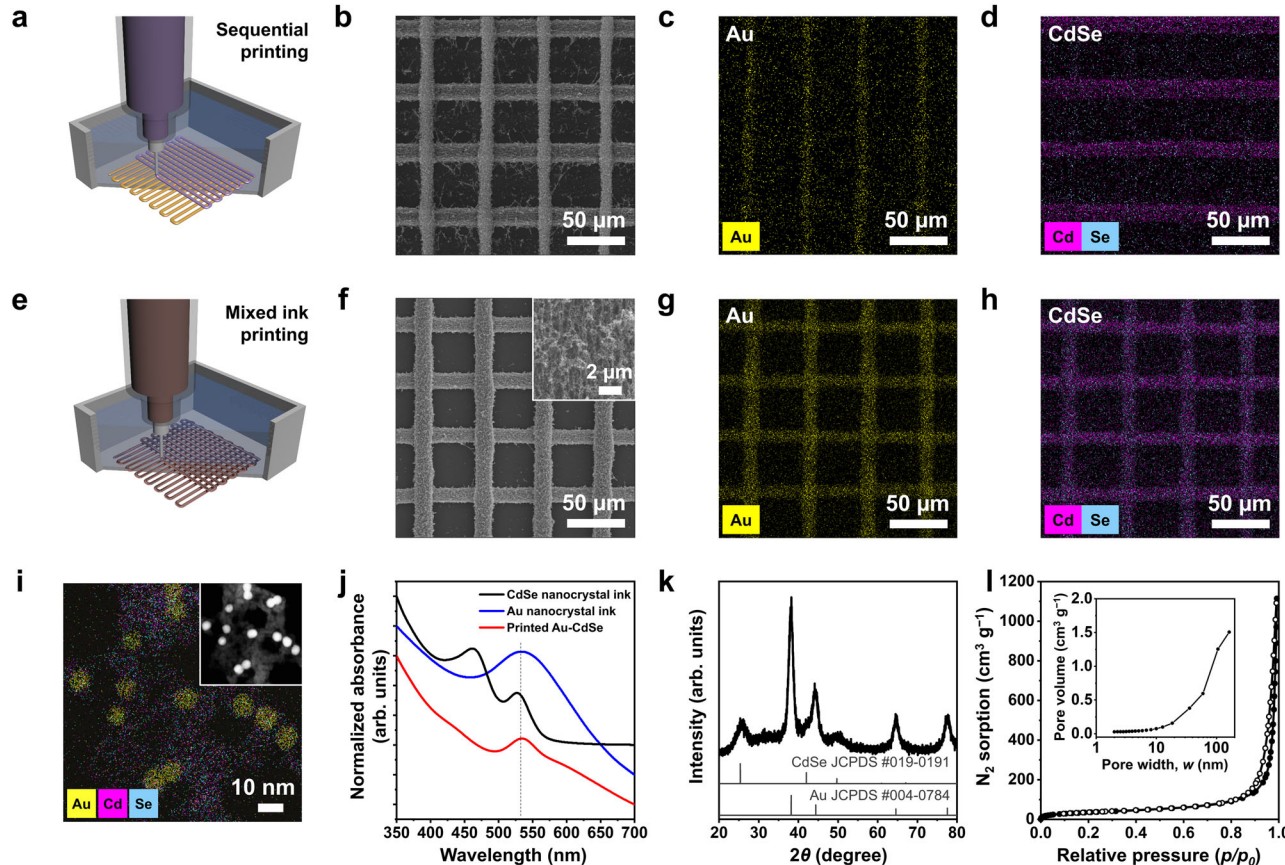

**Fig. 4 | 3D printing of multi-material nanocrystals. a** Scheme for sequential deposition of multiple nanocrystal inks for CdSe and Au nanocrystals. **b** SEM image of sequentially printed Au (vertical) and CdSe (horizontal). EDS mapping image of Au (yellow) (**c**) and Cd (purple) (**d**). **e** Scheme for 3D printing of mixed inks containing CdSe and Au nanocrystals. **f** SEM image of printed Au-CdSe composite. Inset: HRSEM image of printed Au-CdSe. EDS analysis of Au (yellow) (**g**) and Cd (purple) (**h**). **i** Phase analysis of mixed Au-CdSe nanocrystals using HAADF-STEM (inset) and STEM-EDS mapping of Au (yellow), Cd (purple), and Se (blue). **j** UV-Vis absorption spectra of thiomolybdate-capped CdSe (black) and Au (blue) nanocrystal ink, and printed Au-CdSe composite (red). **k** XRD pattern of printed Au-CdSe nanocrystals. The vertical lines indicate the bulk references of CdSe and Au. **i** $N_2$ physisorption isotherm of printed Au-CdSe nanocrystals. Inset: pore size distribution derived from BJH analysis.

CdSe nanocrystals was selected as a model system because metal-semiconductor hybrid systems have broad applications in photonics, optoelectronics, and photocatalysis[56]. A lattice structure with alternating Au and CdSe layers was obtained via the sequential printing of Au and CdSe nanocrystal inks (Fig. 4a). SEM and EDS images showed good separation between the vertical Au and horizontal CdSe layers, without merging at the junctions (Fig. 4b–d). The printed lattice of the mixed Au-CdSe yielded a homogeneous composition of Au and CdSe phases, as confirmed by SEM and EDS images (Fig. 4e–h). The microstructural morphology of the highly porous network was similar to that of single nanocrystal-based samples (Fig. 4f inset). The STEM-EDS maps and high angle annular dark field-scanning TEM (HAADF-STEM) images showed that the Au and CdSe nanocrystals were isolated in the porous multi-branched network (Fig. 4i). The absorption spectrum and XRD pattern confirmed that the Au and CdSe nanocrystals were retained in the composite structure, without atomic-scale structural rearrangements or changes in the electronic spectra (Fig. 4j, k). The BET surface area (132 $m^2$ $g^{-1}$) was similar to the average value of the Au and CdSe printed structures, reflecting its homogeneity (Fig. 4l). These results demonstrate the straightforward 3D printing of multiple materials with multifunctional and synergistic features using our method.

## Discussion
We report the development of a high-precision 3D printing method for the production of sub-10 μm inorganic porous architectures by the ink writing of inorganic nanocrystals in a linker-containing nonsolvent bath. Controlling dispersibility and the subsequent surface linking reaction of the nanocrystals enabled their instant connection during the ink extrusion to produce high-fidelity 3D microarchitectures. This chemical strategy enables the 3D microprinting of a diverse range of functional materials, including metals, semiconductors, magnets, oxides, and so-called "multi-materials", without the use of organic binders or special lithographic equipment. By incorporating low-dimensional materials, such as 1D and 2D materials and molecular clusters, we are confident that our process can be extended for printing an even broader range of inorganic materials with 3D microarchitectures and diverse functionalities.

3D-printed porous materials can be categorised into two distinct classifications: the first being printed lattices (Supplementary Table 9), wherein the porosity emanates from the printed structures, and the second encompassing printed filaments (Supplementary Table 10), inherently characterised by microporosity within the filament itself. Compared with the reports in the former category, e.g., the two-photon based stereolithography-printed lattices, our 3D printing technology, while not exhibiting superior printing resolution, excels remarkably in terms of porosity characteristics, specifically in pore size and surface area. In the latter category, where the focus predominantly centres on 3D printing of graphene-based materials for applications in flexible electronics, our 3D printing technology demonstrates noteworthy advantages in both printing resolutions and the versatility of printable materials. Moreover, our 3D-printed materials conserve the microstructural porosity and properties of the building blocks in the printed

inorganic structures, and will, we suggest, enable their wider implementation in thermal and acoustic insulating, catalytic, electronic, optical, and magnetic components, and in the manufacture of micro-inorganics for energy-storage, microelectronics, micromechanics, and biomedical systems.

## Methods

### Chemicals

Gold(III) chloride trihydrate ($HAuCl_4 \cdot 3H_2O$, 99.9% trace metal basis, Aldrich), chloroplatinic acid hexahydrate ($H_2PtCl_6 \cdot 6H_2O$, ACS reagent 37.5% Pt basis, Aldrich), iron(II) chloride tetrahydrate ($FeCl_2 \cdot 4H_2O$, puriss. p.a. 99%, Aldrich), silver nitrate ($AgNO_3$, 99% ACS reagent, Aldrich), copper(II) acetylacetonate ($Cu(acac)_2$, 99.9% trace metal basis, Aldrich), platinum(II) acetylacetonate ($Pt(acac)_2$, 97%, Aldrich), iron(III) acetylacetonate ($Fe(acac)_3$, 99.9%, Aldrich), cadmium oxide (CdO, 99.99% trace metal basis, Aldrich), selenium powder (Se, 200 mesh 99.999% metal basis, Alfa Aesar), borane tert-butylamine complex (TBAB, 97%, Aldrich), 1,2,3,4-tetrahydronaphthalene (tetralin, 97%, Alfa Aesar), butylamine (99.5%, Aldrich), palmitic acid (99%, Aldrich), 1,2-hexadecanediol (90%, Aldrich), oleic acid (OA, 90%, Aldrich), oleyl amine (OLAm, 70%, Aldrich), trioctylphosphine (TOP, 90%, Aldrich), 1-octadecene (ODE, 90%, Aldrich), dioctyl ether (99%, Aldrich), phenyl ether (99%, Acros organics), ammonium tetrathiomolybdate (ATTM, 99.97%, Aldrich), nitrosyl tetrafluoroborate ($NOBF_4$, 95%, Aldrich), trichloro(1H,1H,2H,2H-perfluorooctyl)silane (PFOCTS, 97%, Aldrich), (3-aminopropyl)triethoxysilane (APTES, 99%, Aldrich), N-methylformamide (NMF, 99%, Aldrich), N,N-dimethylformamide (DMF, 99.8%, Aldrich), hexane (anhydrous 95%, Aldrich), toluene (anhydrous 99.8%, Aldrich), ethyl acetate (EA, 99.5%, SAMCHUN), tetrahydrofuran (THF, 99.5%, SAMCHUN), acetonitrile (anhydrous 99.8%, Sigma), N-methyl-2-pyrrolidone (NMP, anhydrous 99.5%, Sigma), dimethyl sulfoxide (DMSO, 99.9%, Sigma), dichloromethane (DCM, 99.8%, Sigma), chloroform (99.5%, SAMCHUN), pentanol (99%, Sigma), hexanol (98%, Sigma), cyclohexane (99%, Sigma), octane (95%, SAMCHUN), 1-butanol (BtOH, 99%, SAMCHUN), ethanol (99.5%, SAMCHUN), methanol (99.5%, SAMCHUN), 2-propanol (IPA, 99.5%, SAMCHUN), acetone (99.5%, SAMCHUN), and hydrochloric acid (HCl, 37%, Sigma). All the chemicals were used as received, without further purification.

### Synthesis of colloidal nanocrystal inks

**Au nanocrystals.** Au nanocrystals were synthesised using a modified method based on a previous report[57]. 6 nm-sized Au seed nanocrystals were synthesised first. $HAuCl_4$ (0.2 g), tetralin (10 mL), and OLAm (10 mL) were mixed into three-neck round bottom flask at room temperature under $N_2$ flow and vigorous magnetic stirring. TBAB (0.5 mmol), tetralin (1 mL), and OLAm (1 mL) were mixed via sonication for 1 h at room temperature. The solution was then injected into the $HAuCl_4$ solution and stirred for 1 h at room temperature. Acetone was added to the mixture, and centrifuged ($8636 \times g$, 5 min) to collect 6 nm Au seeds. 8 nm Au nanocrystals were synthesised using the synthesised Au seeds. $HAuCl_4$ (0.1 g) was dissolved in ODE (10 mL) and OLAm (10 mL) at room temperature under $N_2$ flow in three-neck round bottom flask. 6 nm Au seeds (30 mg) were added to a solution and the reaction solution was heated to 80 °C for 12 min and kept at this temperature for 2 h. Afterward, the heat source was removed, and the product solution was allowed to cool to room temperature. Ethanol was added to the mixture and centrifuged ($8636 \times g$, 5 min) to remove any unreacted residue. The Au nanocrystals were then dissolved in hexane.

**Ag nanocrystals.** Ag nanocrystals were synthesised using a modified method based on a previous report[58]. $AgNO_3$ (1.7 g), $Cu(acac)_2$ (0.2 g), and toluene (3.468 mL) were mixed into three-neck round bottom flask at room temperature. Butylamine (1.352 mL) and palmitic acid (0.5 g) were added to this mixed solution. The reaction mixture was heated to 110 °C for 17 min and kept at this temperature for 2 h. Afterward, the heat source was removed, and the product solution was allowed to cool to room temperature. Methanol was added to the mixture, and centrifuged ($8636 \times g$, 5 min) to remove any unreacted residue. The Ag nanocrystals were then dissolved in hexane.

**FePt nanocrystals.** FePt nanocrystals were synthesised using a modified method based on a previous report[59]. $Pt(acac)_2$ (0.196 g), $Fe(acac)_3$ (0.177 g), reducing agent 1,2-hexadecanediol (1.292 g), OA (0.16 mL), OLAm (0.164 mL), and dioctyl ether (20 mL) were mixed into three-neck round bottom flask at room temperature in glove box. The reaction mixture was heated to reflux at 286 °C and kept at this temperature for 30 min. Afterward, the heat source was removed, and the product solution was allowed to cool to room temperature. Ethanol was added to the mixture and centrifuged ($8636 \times g$, 5 min) to remove any unreacted residue. The FePt nanocrystals were then dissolved in hexane.

**CdSe nanocrystals.** CdSe nanocrystals were synthesised using a modified method based on a previous report[60]. CdO (0.255 g), OA (3.11 mL), and ODE (35 mL) were mixed into three-neck round bottom flask at room temperature in glove box. The solution was heated to 180 °C for 31 min and kept at this temperature for 1 h to form a clear solution under $N_2$ atmosphere. Subsequently, the solution was heated up to 250 °C for 14 min. TOP-Se solution of Se (0.051 g), TOP (0.3 mL), and ODE (5 mL) were mixed and stirred more than 1 h in glove box. The TOP-Se solution was then injected into the CdO solution and stirred for 2 min. Afterward, the heat source was removed, and the product solution was allowed to cool to room temperature. Methanol, acetone, IPA were added to the mixture and centrifuged ($8636 \times g$, 5 min) to remove any unreacted residue. The CdSe nanocrystals were then dissolved in hexane.

**$Fe_3O_4$ nanocrystals.** $Fe_3O_4$ nanocrystals were synthesised using a modified method based on a previous report[61]. $Fe(acac)_3$ (0.7063 g), 1,2-hexadecanediol (2.5844 g), OA (1.91 mL), OLAm (1.975 mL), and phenyl ether (20 mL) were mixed into three-neck round bottom flask at room temperature in glove box. The reaction mixture was heated to 200 °C for 35 min under flow of $N_2$, and then kept at this temperature for 30 min. Subsequently, the mixture was heated to 265 °C for 13 min, then kept at this temperature for 30 min. The black-brown mixture was cooled to room temperature by removing the heat source. Ethanol was added to the mixture, and centrifuged ($6909 \times g$, 5 min) to remove any unreacted residue. The $Fe_3O_4$ nanocrystals were dissolved in hexane.

**Synthesis of inorganic ligands-capped nanocrystal inks.** Negative charges on the surfaces of nanocrystals were introduced by the ligand exchange process. All ligand exchange reactions were performed in a $N_2$-filled glovebox using a typical two-phase ligand-exchange strategy[62]. ATTM was exploited as an inorganic ligand to replace the existing organic ligands. ATTM (0.9 g) was dissolved in NMF (30 mL) and stirred for 10 min. Then, 10 mL of nanocrystals containing hexane solution (30 mg mL⁻¹) was added to a vial containing 30 mL of ATTM solution (30 mg mL⁻¹). The mixture was vigorously stirred until the phase transfer of nanocrystals from the upper hexane phase to the bottom NMF phase was completed. After ligand exchange, the upper hexane phase was discarded, and the bottom layer of ATTM-capped nanocrystals was collected by the addition of IPA (210 mL). The purification step was repeated two times to collect the ATTM-capped nanocrystals.

For the case of $Fe_3O_4$ nanocrystals, the two-phase ligand exchange process cannot be adopted directly. The ligand stripping process was conducted first to remove the existing organic ligands[63]. $NOBF_4$ (0.9 g) was dissolved in DMF (30 mL) and stirred for 10 min. To form an immiscible two-phase mixture, 10 mL of nanocrystals in hexane (30 mg mL⁻¹) was added to a vial containing 30 mL of $NOBF_4$ solution (30 mg mL⁻¹). The immiscible two-phase mixture was vigorously

stirred until the phase transfer of nanocrystals from the upper hexane phase to the bottom NOBF$_4$ phase is completed. After ligand stripping, the upper hexane phase was discarded, and the bottom solution was collected by the addition of toluene (210 mL). The purification step was repeated two times to remove unreacted NOBF$_4$ species. The stripped nanocrystals were redispersed in 30 mL of ATTM stock solution (30 mg mL$^{-1}$) and stirred overnight. The solution was precipitated by the addition of IPA (210 mL) to collect the ATTM-capped Fe$_3$O$_4$ nanocrystals.

The ATTM- capped nanocrystals were dispersed in NMF to form stable colloidal solution with concentration of 25 mg mL$^{-1}$ (Ag), 50 mg mL$^{-1}$ (Au), 45 mg mL$^{-1}$ (FePt), 50 mg mL$^{-1}$ (CdSe), and 70 mg mL$^{-1}$ (Fe$_3$O$_4$). A colloidal solution of ATTM-capped nanocrystals was exploited as 3D printing ink without any organic additives.

## Hansen solubility parameter(HSP) difference ($R_a$) calculations

The HSPs indicate the cohesive energy density of a chemical resulting from the interactions of a given solvent molecule[64]. The energy needed to break all the cohesive bonds involves dispersion force, permanent dipole-permanent dipole forces, and hydrogen bonding. Thus, the total solubility parameter can be calculated in Eq. 1

$$\delta_t^2 = \delta_d^2 + \delta_p^2 + \delta_h^2 \qquad (1)$$

Where $\delta_t$ (MPa$^{1/2}$) is the solubility parameter and $\delta_d$, $\delta_p$, and $\delta_h$ are the dispersion force, dipole interaction force, and hydrogen bonding force term, respectively. $R_a$ is the difference between the HSPs of two materials, given by Eq. 2

$$R_a^2 = \{4(\delta_{d1} - \delta_{d2})^2 + (\delta_{p1} - \delta_{p2})^2 + (\delta_{h1} - \delta_{h2})^2\} \qquad (2)$$

A smaller $R_a$ indicates that the HSP of the two materials are likely to be miscible.

Dielectric constants of solvents were obtained in the ref. 65.

## Wet 3D microprinting process of inorganic nanocrystals

**Preparation of micronozzles, substrates, and linker containing solidification baths.** The borosilicate glass capillaries were cleaned by rinsing with methanol, acetone, and IPA under sonication for 5 min each. The borosilicate glass capillaries were pulled to prepare a nozzle with a pipette puller (P-1000, Sutter Instruments). The pipette-pulling parameters, such as heat, pull, velocity, time, delay, and pressure, were tuned to fabricate diameter- and morphology-controlled glass pipettes. The pre-pulled glass pipettes were O$_2$-plasma treated for surface hydrophilisation. Hydrophobic surface treatment was performed by a PFOCTS self-assembled monolayer (SAM) deposition step using a vapour-phase technique. The trichlorosilane-based head groups reacted with the hydroxyl group on the substrate to form a stable covalent bond. The PFOCTS SAM-coated glass pipettes were thermally treated at 120 °C for 20 min and rinsed with hexane to remove the unreacted PFOCTS species. The Si wafers were cleaned by rinsing with methanol, acetone, and IPA under sonication for 5 min each. The clean Si substrates were O$_2$-plasma treated for surface hydrophilisation. Hydrophilic surface treatment was performed via APTES SAM deposition using a vapour-phase technique. The triethoxysilane-based head groups reacted with the hydroxyl group on the substrate to form a stable covalent bond. The APTES SAM-coated substrates were thermally treated at 100 °C for 30 min and rinsed with toluene to remove the unreacted APTES species. Finally, the linker baths were prepared by dissolving HAuCl$_4$, H$_2$PtCl$_6$, and FeCl$_2$·4H$_2$O in 1-butanol to obtain 0.5–1 mM solution.

**Wet 3D microprinting procedure.** The ink formulation condition was given in the Supplementary Table 11. The printing machine consisted of a micronozzle connected to a dispenser (Ultimus 2, Nordson EFD) and a

three-axis ($x$,$y$,$z$) stepping motor nanostage (Aerotech). The inorganic ligand-capped nanocrystals were loaded into a PFOCTS SAM-coated micronozzle, and the inks were pneumatically driven through a micronozzle at 2.1–100 kPa. The APTES SAM-coated Si substrate was attached to a glass petri dish and placed on a three-axis ($x$,$y$,$z$) nanostage. The distance between the micronozzle and the substrate was fixed at 10 μm. After controlling the distance, the linker bath solution was poured into a glass petri dish. Their positions and moving speeds were accurately controlled in real time using a motion composer software (A3200, Aerotech). The stage was translated at a speed of 1.2 mm s$^{-1}$ during printing and the overall process was monitored using a side-view charge coupled device (CCD) camera (MicroPublisher 5.0 RTV, QImaging). Also, the movies showing the printing process were recorded using a CCD camera (Supplementary Movie 1–3).

**Supercritical drying.** The printed object was solvent-exchanged with fresh butanol at 25 °C several times. After a complete exchange of the solvent, the printed wet state objects were transferred into a supercritical fluid extractor (SFT-110XW, Supercritical Fluid Technologies Inc.) with an excess amount of butanol to prevent the evaporation of the solvents. The chamber was flushed with liquid CO$_2$ to exchange the butanol. To convert the liquid CO$_2$ to the supercritical state, the vessel was pressurised and heated to 1800 psi and 60 °C, respectively. The butanol-CO$_2$ mixture was extracted continuously through the exit of the vessel until all solvents were removed.

**Post-chemical treatment for surface linker and ligand removal.** The printed object was exposed into 0.1 M HCl solution for 1 h. Then printed wet state object was solvent-exchanged with fresh ethanol at 25 °C several times. The ethanol-CO$_2$ mixture was extracted continuously through the supercritical drying process.

**Calculation of mass fractal dimension.** The calculation of the 2D fractal dimension ($d_p$) involves the following steps: (a) Thresholding: the original TEM images are subjected to thresholding to obtain binary images. (b) Image processing: various image processing techniques are applied to the binary images to reduce noise and preserve the sharp edges of cluster objects. (c) Measurement: the area ($A$) and perimeter ($P$) of each component in the binary image are measured. (d) Curve fitting: a linear curve of log($P^2$) vs. log($A$), where the slope represents the 2D fractal dimension $d_p$ of the image.

The image processing mentioned in step (b) involves several sequential steps, including several filters and other operations.

(1) Opening Operation: This operation removes small bright regions while preserving sharp edges. We utilise a disk matrix with a radius of 3 as the structural element.

(2) Closing Operation: This operation removes small dark regions while preserving sharp edges.

   Subsequently, we employ two edge-preserving filters

(3) Bilateral Filter: This filter smooths the images while preserving strong edges.

(4) Guided Filter: This filter smooths the image while preserving local structure.

   Additionally, two denoising filters are utilised:

(5) Total Variation Filter: This filter reduces noise while preserving edges and textures.

(6) Nonlocal Means Filter: This filter removes noise by exploiting self-similarity in the image.

   The following steps complete the image processing:

(7) Removal of objects touching the edges of the image.

(8) Hole Filling Operation: This step fills in the holes inside a cluster object.

Furthermore, the thresholding process in step (a) employs the method of inter-class variance maximisation on the original TEM

images. It is important to note that thresholding should be performed multiple times to ensure that the resulting binary image maintains its binary nature.

The 3D fractal dimension $d_f$ can be computed using Eq. (3), which is derived from the morphological aggregation model with a variable compactness parameter[66–68]. Equation 3 offers superior accuracy compared to the previously proposed equation and is applicable to spherical primary particles within the validity region[66].

$$d_f = 6.03 \cdot \left(d_p\right)^3 - 25.63 \cdot \left(d_p\right)^2 + 34.13 \cdot d_p - 11.55 \quad (3)$$

In our calculation, for different pressures, the area and perimeter data obtained from 5 TEM images are used collectively. We use Wolfram Mathematica to do the all image processing mentioned above.

## Characterisations

**Microscopy analyses.** The dimension and microstructure of the printed inorganic nanocrystal-based porous materials were imaged using OM and SEM, respectively. Optical imaging was performed using an OM (BX51M, Olympus). SEM (including tilted views) and EDS mapping image were collected using a field-effect SEM (Nova NanoSEM, FEI and SU7000, Hitachi High-Tech) with a 10 kV (SEM image) and 20 kV (EDS mapping image) electron beam. The CCD images and movies were obtained using a CCD camera (MicroPublisher 5.0 RTV, QImaging). The TEM images were obtained at 200 kV using a JEOL-2100 microscope (JEOL). HR-TEM, HAADF-STEM imaging, and spectral imaging based on STEM-EDS were performed at 200 kV using a JEM-2100F (JEOL) and JEM-ARM300F microscope (JEOL). For analysis, the printed objects were crushed and suspended in methanol by ultrasound for 15–120 s, depending on their dispersing ability. The cross-sectional TEM sample of Ag filament was prepared using a focused ion beam (FEI Helios NanoLab 450). Bright-field transmission electron microscopy (BF-TEM) and scanning TEM images were obtained using a chromatic aberration (Cs) corrected TEM (Grand ARM300F, JEOL) at an acceleration voltage of 160 kV.

**$N_2$ adsorption/desorption analysis.** The sample porosities were determined using an $N_2$ sorption analyser (BELSORP-Max, BEL) operated at 77 K. Prior to the measurements, the sample surfaces were evacuated at 70 °C for 12 h under vacuum conditions to clean the surfaces. The specific surface areas of the samples were calculated using the BET equation, while their pore size distributions were derived from the adsorption branches of the isotherms using the BJH (Barrett-Joyner-Halenda) method. The silica equivalent surface areas were calculated by the relative density method[69]. Here, the density of silica is assumed to be an average density of quartz (2.65 mg cm$^{-3}$), tridymite (2.31 mg cm$^{-3}$), and cristobalite (2.33 mg cm$^{-3}$), 2.43 mg cm$^{-3}$.

**X-ray diffraction analysis.** The XRD patterns were obtained using a high-power X-ray diffractometer (D/MAX2500V/PC, Rigaku) equipped with Cu Kα radiation and operated at 40 kV and 200 mA.

**ζ-potential analysis.** The ζ-potential data were collected using a Zetasizer Nano ZS instrument (Malvern). Inorganic ligand-capped Ag nanocrystals were measured before and after exposure to $Au^{3+}$, $Pt^{4+}$, and $Fe^{2+}$ linker solution, respectively.

**X-ray photoelectron spectroscopy.** XPS spectra were acquired using an X-ray photoelectron spectrometer (ESCALAB 250XI, Thermo Fisher Scientific) with a monochromatic Al Kα X-ray source (1486.6 eV). All XPS spectra were corrected with adventitious C 1 s peak at 284.8 eV. For analysis, inorganic ligand-capped Ag nanocrystals were printed on the Si substrate filled with butanol, $Au^{3+}$, $Pt^{4+}$, and $Fe^{2+}$ linker solutions,

respectively. After drying with a supercritical fluid, all the samples were kept in a glove box to prevent oxidation before analysis.

**Thermal stability analysis.** The thermal stability of the printed Au nanocrystal-based porous material was investigated by thermogravimetric analysis (TGA Q500, TA Instruments) in the temperature range of 25–700 °C at a heating rate of 10 °C min$^{-1}$ under a nitrogen atmosphere.

**Magnetic property measurement.** The magnetic property was measured by using a superconducting quantum interference device-vibrating sample magnetometer (SQUID-VSM, Quantum Design).

**Optical property measurement.** The UV-vis absorption spectra were measured using a UV-vis spectrophotometer (UV-2600, Shimadzu). The photoluminescence spectra were investigated using a UV-Vis-NIR spectrophotometer (Cary 5000) at room temperature. For analysis, wet state CdSe was prepared by extruding inorganic ligand-capped CdSe nanocrystals (1 mL) into the 0.5 mM concentration of $Fe^{2+}$ linker solution (50 mL). The subsequent supercritical drying process can produce the dried state CdSe powder. The obtained dried state CdSe was suspended in NMF by ultrasound for 120 s. To investigate the specific oxidation effect on the optical properties, CdSe wet state and dried state dispersions were treated under UV irradiation under the air atmosphere for 24 h.

**Electrical property measurement.** The electrical conductivity was measured by a four-point van der Pauw method (Keithley 2400 multimeter controlled by Lab trace 2.0 software, Keithley Instrument, Inc.). For analysis, inorganic ligand-capped Au nanocrystal inks (5 μl) were casted on the Si substrate filled with 0.5 mM $Au^{3+}$ linker solution (2 mL) followed by supercritical drying process to produce the Au aerogel film. For the preparation of xerogel films, the solvent was evaporated in ambient conditions. After complete evaporation of the solvent, a homogeneous film was formed on the substrate. The Au xerogel film was heat treated at 600 °C for 1 h with a ramping rate of 5 °C min$^{-1}$ under $H_2$ (99.999%) in a tube furnace.

**Materials for electrochemical measurement.** Electrochemical measurements were conducted in 0.1 M $HClO_4$ (70 wt%, Aldrich) electrolyte, using a three-electrode electrochemical cell on a potentiostat (Autolab PGSTAT204 Electrochemical Workstations) and a rotator (Autolab) at room temperature. A rotating disk electrode (RDE, Autolab) with a glassy carbon disk (0.1963 cm$^2$) was used as a working electrode. An Ag/AgCl (Autolab; saturated KCl filling solution) and a glassy carbon rod (Autolab) were used as the reference and counter electrodes, respectively. Before use, the Ag/AgCl reference electrode was calibrated with respect to the reversible hydrogen electrode (RHE). RHE calibration was performed in an $H_2$-saturated 0.1 M $HClO_4$ solution with a Pt sheet as working electrode and Ag/AgCl as the reference electrode. With continuous $H_2$ bubbling, a stable open circuit potential was obtained within 20 min, corresponding to the RHE conversion value.

**Working electrode preparation.** The working electrode was prepared by printing wet-state FePt onto the glassy carbon disk of an RDE. The subsequent acid treatment (0.1 M HCl) and supercritical drying process can produce the dried state FePt film. The Pt loading was controlled at 50 μg$_{Pt}$ cm$^{-2}$. The electrochemical properties of this FePt film were compared with two benchmark catalysts: commercial 20 wt% Pt/C (Hispec3000, Johnson Matthey) as well as FePt/C sample (20 wt% Pt) synthesised by modifying previously reported method[70]. For the benchmark catalysts, 5 mg of catalyst powder were mixed with DI water (200 μL), 5 wt% Nafion solution (40 μL, in isopropyl alcohol, Aldrich), and isopropyl alcohol (960 μL, 99.5%, Aldrich) and ultrasonicated for

30 min to prepare catalyst inks. The catalyst inks were drop-casted onto the glassy carbon disk of an RDE and dried at room temperature to form catalyst films. The Pt loadings for Pt/C and FePt/C were controlled at $20 \mu g_{Pt} cm^{-2}$.

**Electrocatalytic property measurement.** Prior to electrochemical measurements, the prepared catalyst layers were activated in an Ar-saturated electrolyte to form clean catalytic surfaces. The printed FePt catalyst was activated by cycling the potential between 1.0 and 1.5 V (vs. RHE) for 2000 cycles at a scan rate of $500 mV s^{-1}$ while the benchmark catalysts were cleaned by 100 potential cycles between 0.05 and 1.0 V (vs. RHE) at a scan rate of $500 mV s^{-1}$. After the electrochemical activation, cyclic voltammogram (CV) curves were obtained at the potential between 0.05 and 1.0 V (vs. RHE) at a scan rate of $100 mV s^{-1}$. The electrochemically active surface areas (ECSAs) of catalysts were estimated using the integrated charge from the underpotentially deposited hydrogen ($H_{upd}$) peak between 0.05 and 0.35 V (vs. RHE).

ORR activities of catalyst layers were measured from linear sweep voltammetry (LSV) curves using a potential sweep from −0.01 to 1.10 V (vs. RHE) in an $O_2$-saturated 0.1 M $HClO_4$ solution at a rotation speed of 1600 rpm and a scan rate of $20 mV s^{-1}$. The ORR activities of catalysts were presented after *iR*-drop correction and background removal. Solution resistance was determined at the X-intercept at the high-frequency region of the Nyquist plot, obtained by measuring EIS at 0.70 V (vs. RHE) with an AC potential amplitude of 10 mV from 10,000 to 1 Hz. ORR kinetic current densities of catalysts calculated using the Koutecký−Levich equation. ORR mass activities (MAs) were evaluated at 0.9 V (vs. RHE) by normalising kinetic current density with catalyst loading. ORR accelerated durability tests (ADTs) were conducted in an Ar-saturated 0.1 M $HClO_4$ solution in a potential range from 0.6 to 1.0 V (vs. RHE) at a scan rate of $50 mV s^{-1}$ for 10,000 cycles. After cycling, the ORR activity was measured in fresh 0.1 M $HClO_4$. The ORR polarisation curves before and after potential cycling were obtained under the same conditions as for the above ORR measurement.

## Data availability
The data that supports the findings of the study are included in the main text and supplementary information files. The source data underlying the figures of the main text are provided within the "Source Data" file. All raw data generated during the current study are available from the corresponding authors upon request. Source data are provided with this paper.

## Code availability
The Wolfram Mathematica codes used in this study are outlined in the method section and available from the corresponding authors.

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

## Acknowledgements

We acknowledge the Nano·Material Technology Development Program (NRF-2018M3A7B8060697), the mid-career researcher program (NRF-2022R1A2C3009129 and NRF-2021R1A2C2007495), and the Creative Materials Discovery Program (NRF-2020M3D1A1110502) through the National Research Foundation of the Republic of Korea (NRF) funded by Ministry of Science and ICT. This work was supported by the Korea Institute Energy Technology Evaluation and Planning (KETEP) grant funded by the Korea government (MOTIE) (no. 20213030030260). The work was supported by the program of Future Hydrogen Original Technology Development (NRF-2021M3I3A1082879) through the NRF funded by the Korea government (Ministry of Science and ICT). This work was supported by IBS-R019-D1.

## Author contributions

M.S., J.S.S., J.Y.K. designed the experiments, analysed the data, and wrote the paper. M.S., Y.K., B.V.C., S.E.Y., S.H.H., D.H.G., M.K., H.Y.J. and R.S.R. carried out the synthesis and basic characterisation of materials. M.S., D.S.B., J.S.L. and S.H.J. performed the characterisation of porosity. H.Y.K. and J.Y.K. carried out the measurement of the electrocatalytic activity. H.L. performed the calculation of the mass fractal dimension. S.L. and J.-W.Y. performed the characterisation of magnetic properties. All authors discussed the results and edited and commented on the manuscript.

## Competing interests

Ulsan National Institute of Science and Technology (UNIST) has filed a patent, PCT/KR2022/003817 (inventors: M.S., Y.K., and J.S.S.) that covers the wet 3D microprinting chemistry and methods reported in this article. All other authors declare no competing interests.
