## [Peer Review File · Nature Communications]

3D microprinting of inorganic porous materials by chemical linking-induced solidification of nanocrystalsEditorial Note: This manuscript has been previously reviewed at another journal that is not operating a transparent peer review scheme. This document only contains reviewer comments and rebuttal letters for versions considered at *Nature Communications*.

REVIEWERS' COMMENTS

Reviewer #1 (Remarks to the Author):

The authors have satisfactorily addressed the concerns raised from previous rounds of reviewers. I would like to recommend publication of this manuscript in Nature Communications.

Reviewer #2 (Remarks to the Author):

Based on the past reviews and subsequent edits, the paper is of sufficient quality for publication. The authors also clarified the relationship between the present results and prior reports in the literature, which will help readers contextualize the claims made.

Response to the reviewers' comments

The followings are the responses to the reviewers' comments for the manuscript "3D microprinting of inorganic porous materials by chemical linking-induced solidification of nanocrystals."

▪ Reviewer #1

Comment 1: The authors have satisfactorily addressed the concerns raised from previous rounds of reviewers. I would like to recommend publication of this manuscript in Nature Communications.

Response: We sincerely appreciate the reviewer's positive comment on our manuscript.

▪ Reviewer #2

Comment 1: Based on the past reviews and subsequent edits, the paper is of sufficient quality for publication. The authors also clarified the relationship between the present results and prior reports in the literature, which will help readers contextualize the claims made.

Response: We sincerely appreciate the reviewer's positive comment on our manuscript.